# Discovering Potential Anti-Oral Squamous Cell Carcinoma Mechanisms from Kochiae Fructus Using Network-Based Pharmacology Analysis and Experimental Validation

**DOI:** 10.3390/life13061300

**Published:** 2023-05-31

**Authors:** Youn-Sook Kim, Jin-Choon Lee, Minhyung Lee, Hae-Jin Oh, Won G. An, Eui-Suk Sung

**Affiliations:** 1Research Institute for Longevity and Well-Being, Pusan National University, Busan 46241, Republic of Korea; younskim@pusan.ac.kr; 2Department of Otolaryngology-Head and Neck Surgery, College of Medicine, Pusan National University and Research Institute for Convergence of Biomedical Science and Technology, Pusan National University Yangsan Hospital, Yangsan 50612, Republic of Korea; lic0209@hanmail.net (J.-C.L.);; 3Department of Pharmacology, School of Korean Medicine, Pusan National University, Yangsan 50612, Republic of Korea

**Keywords:** natural product, oral squamous cell carcinoma, Kochiae Fructus, network-based pharmacological analysis, molecular docking analysis, autophagy, oleanolic acid

## Abstract

The natural product Kochiae Fructus (KF) is the ripe fruit of *Kochia scoparia* (L.) Schrad and is renowned for its anti-inflammatory, anticancer, anti-fungal, and anti-pruritic effects. This study examined the anticancer effect of components of KF to assess its potential as an adjuvant for cancer treatment. Network-based pharmacological and docking analyses of KF found associations with oral squamous cell carcinoma. The molecular docking of oleanolic acid (OA) with LC3 and SQSTM1 had high binding scores, and hydrogen binding with amino acids of the receptors suggests that OA is involved in autophagy, rather than the apoptosis pathway. For experimental validation, we exposed SCC-15 squamous carcinoma cells derived from a human tongue lesion to KF extract (KFE), OA, and cisplatin. The KFE caused SCC-15 cell death, and induced an accumulation of the autophagy marker proteins LC3 and p62/SQSTM1. The novelty of this study lies in the discovery that the change in autophagy protein levels can be related to the regulatory death of SCC-15 cells. These findings suggest that KF is a promising candidate for future studies to provide insight into the role of autophagy in cancer cells and advance our understanding of cancer prevention and treatment.

## 1. Introduction

The development of advanced treatments for patients with fatal diseases has been a major goal for researchers. Although modern treatments can lead to successful remission during the early stages of cancer, the treatment of advanced cancers remains problematic and patients often suffer from undesired side effects. The oral cavity is vulnerable to viruses and carcinogens. The most common site of intraoral cancer is the tongue. GLOBOCAN estimated in 2020 that there were approximately 377,713 new cases and 177,757 new deaths from lip and oral cavity cancer worldwide, making them the most prevalent head and neck squamous cell carcinomas (HNSCCs) [1]. Since prophylactic treatment is difficult, early molecular diagnosis and treatment are very important for the prevention, prediction, and treatment of oral cavity cancer before aggressive progression occurs.

Typically, chemoradiotherapy (CRT) is performed after surgery. For CRT, cisplatin (CP), an alkylating agent [2], is generally used as a radiation sensitizer [3]. CP is the most important anticancer agent in the treatment of head and neck tumors, and is typically used at doses of 80–100 mg/m^2^ every 3–4 weeks. The response rate is 14–41%, but no proportional relationship between the dose and response rate has been confirmed [4]. CP is nephrotoxic, so caution is required during its administration.

Expectations for natural products as anticancer adjuvants that can reduce the risk of the side effects and drug resistance associated with synthetic drugs are growing, and the herbal resources for research are limitless [5,6]. Natural products can have positive effects on the human body by acting on genes, or indirectly on conjugates that affect metabolic pathways; historically, they have played an important role in combating cancers [7,8].

Dried ripe Belvedere fruit [*Kochiae scoparia* (L.) Schrad] is known as Kochiae Fructus (KF). For over 2000 years, KF has been used to treat diseases [9]. KF contains terpenoids, flavonoids, carbohydrates, essential oils, and amino acids [10]. Recent studies have demonstrated anti-inflammatory [11], anticancer [12], anti-fungal [13], and anti-pruritic [14] effects of KF extract (KFE) and compounds in KF. Terpenoids such as momordin Ic induces HepG2 cell apoptosis [15] and flavonoids such as hispidulin has been found to have significant anticancer effects [16].

Multiple components of KF and their various targets act on biological processes at many levels. However, there have been no comprehensive studies of the biological effects of components of KF and their gene targets. To study the interactions among compounds, targets, and diseases at a systems level and identify key genes and functions in biological processes, we analyzed KF using an integrated computational approach called network-based pharmacological analysis. We used various biological databases and tools to identify components, predict target genes, and perform functional enrichment analysis thereof [17]. We used the bioinformatics tool Metascape [18] to perform a comprehensive gene annotation and functional enrichment analysis of the KF gene list. Molecular docking analysis was performed with AutoDock Vina to predict the binding and affinity of the target proteins and compounds.

In this study, we considered the possibility of using KF for the treatment of oral squamous cell carcinoma (OSCC). We focused on the involvement of KF targets (KFTs) in squamous carcinoma cell apoptosis and autophagy signaling pathways; the SCC-15 and SCC-25 squamous carcinoma cells used were derived from a human tongue cancer. The KFE and oleanolic acid (OA) were subjected to experimental verification to determine their anticancer effects on SCC-15. The additional SCC-25 cell line was only used to confirm noticeable results relevant to KFE and OA autophagy inducement. The results of this study constitute bioinformatic resources for further studies of the anticancer potential of KF and its active compounds. The data may provide a basis for further research to elucidate the mechanisms underlying biomolecular actions.

## 2. Materials and Methods

### 2.1. KF Compounds

KF compounds were identified in four databases: SymMap v2 (http://www.symmap.org/, accessed on 11 February 2023) [19], TCMSP (https://tcmsp-e.com/index.php, accessed on 11 February 2023) [20], HIT 2.0 (http://www.badd-cao.net:2345, accessed on 11 February 2023) [21], and HERB (http://herb.ac.cn/, accessed on 11 February 2023) [22]. Venn diagrams were used to visualize the compounds (https://bioinformatics.psb.ugent.be/webtools/Venn/, accessed on 11 February 2023). The structures of the compounds were elucidated using ChemDraw 20.0 and the 3D structures were visualized by PyMOL™ 2.5.4 (license expires 12 December 2023). To identify active compounds, the TCMSP database was searched using the following absorption, distribution, metabolism, and excretion (ADME) parameters: ‘molecular weight (MW, 180–500 Daltons)’, ‘oral bioavailability (OB ≥ 20%)’ [23], ‘drug-likeness (DL ≥ 0.18)’ [24], and ‘Caco2 permeability (Caco2 ≥ −0.04)’ [25]. Additional parameters included Lipinski’s rule of five (Ro5) [26], a bioavailability score (BAC) ≥ 0.55, high gastrointestinal absorption, and bioavailability (according to a radar plot created with SwissADME; http://www.swissadme.ch/, accessed on 11 February 2023) [27].

### 2.2. Target Gene Identification

KFTs were identified in the TCMSP, SymMap, HIT 2.0, HERB, and Similarity Ensemble Approach (SEA; http://sea.bkslab.org/, accessed on 11 February 2023) databases [28]. To identify potential targets of each compound in the SEA database, the maximum Tanimoto coefficient was set to ≥0.4 and the *p*-value was set to ≤1.0 × 10^−16^. Official gene names were identified in the UniProt database (http://www.uniprot.org, accessed on 11 February 2023) [29]. Genes targeting OSCC were identified in the DISEASE network of the STRING database in Cytoscape 3.9.1. (https://cytoscape.org/, accessed on 11 February 2023). Genes associated with autophagy and pathways were identified in Kyoto Encyclopedia of Genes and Genomes (KEGG) pathways (https://www.genome.jp/kegg/pathway.html, accessed on 11 February 2023). Among KFTs, common genes related to OSCC, apoptosis, and autophagy were identified using a FunRich 3.1.3 Venn diagram.

### 2.3. Functional Enrichment Analysis, Protein–Protein Interaction (PPI) Network Construction, and Screening of Hub KFTs

The KFT gene list was uploaded to Metascape (https://metascape.org, accessed on 11 February 2023) for enrichment analysis [16]. Metascape provides data on enrichment pathways, PPI network structures, and gene-set annotation functions using >40 biological databases. PPI networks for the targets were constructed using Cytoscape 3.9.1 and the STRING 11.5 database.

Hub KFTs and OA target genes in PPI networks were selected by ranking them according to the maximal clique centrality (MCC) values calculated by the cytoHubba plugin in Cytoscape. The MCC score, which ranges from −1 to 1, indicates the centrality of nodes according to density (i.e., relative importance) in the network.

### 2.4. Molecular Docking Analysis Using AutoDock Vina

To assess molecular docking, PyMOL was used to visualize the protein, ligand, and protein–ligand complex. The PDB IDs of proteins were obtained from the RCSB PDB database (https://www.rcsb.org/, accessed on 15 February 2023). The PDB structure of the ligand was downloaded from PubChem (https://pubchem.ncbi.nlm.nih.gov/, accessed on 15 February 2023). Using AutoDockTools-1.5.7., the PDB files of proteins, ligands, and protein–ligand complexes were converted to PDBQT files, and a grid box for the docking region of the protein was drawn. Protein–ligand binding conformation and binding affinity were determined using AutoDock Vina (https://github.com/ccsb-scripps/AutoDock-Vina, accessed on 15 February 2023) [30,31]. LigPlot+ (version 2.2.8) was used to graph the protein–ligand interaction [32].

### 2.5. Preparation of KFE

KF was obtained from Kwangmyungdang Medical Herbs (Ulsan, Republic of Korea). Ground KF (100 g) was soaked in 1 L of methanol. After sonication for 30 min, the mixture was kept at room temperature for 48 h. The supernatant was collected and filtered (No. 20 Whatman; GE Healthcare, Chalfont, UK). To derive a lyophilized powder from the filtrate, evaporation using a vacuum evaporator (Eyela; Tokyo Rikakikai, Tokyo, Japan) and lyophilization using a freezer dryer (FreeZone 6 Liter Console Freeze Dry System; Labconco, Kansas City, MO, USA) were performed sequentially. Finally, 4.5 g of powder was obtained for a yield of 4.45%. The powder was dissolved in dimethyl sulfoxide (DMSO) and applied to the cells.

### 2.6. Cell Culture

SCC-15 and SCC-25, both human squamous cell carcinoma cell lines, were purchased from ATCC (Manassas, VA, USA) and maintained in minimal essential medium (MEM; Corning, Manassas, VA, USA) containing 10% fetal bovine serum (FBS) (*v*/*v*) and 1% (*v*/*v*) penicillin/streptomycin (Gibco, Grand Island, NY, USA) at 37 °C in a CO_2_ incubator (Thermo Fisher Scientific, Cincinnati, OH, USA).

### 2.7. Cell Viability Assay

SCC-15 cell viability following treatment with KFE was measured using an MTT assay. Cells were seeded in a 96-well plate (1 × 10^4^ cells/well) and incubated for 24 h at 37 °C in an incubator with a 5% CO_2_ atmosphere. Various concentrations (0, 50, 75, and 125 μg/mL) of KFE were applied to the wells and the cells were incubated for 24 h. Then, the cells were stained with 0.5 mg/mL MTT for 4 h at 37 °C, while shielded from light. The medium was removed and 200 μL of DMSO was added. The absorbance was detected at 570 nm using an ELISA microplate analyzer (SpectraMax iD3; Molecular Devices, San Jose, CA, USA). Cell viability following treatment with OA (O5504; Sigma-Aldrich, St. Louis, MO, USA) and CP (P4394; Sigma-Aldrich) was confirmed using the Cell Counting-Kit-8 (CCK-8) assay (Dojindo Molecular Technologies, Rockville, MD, USA). SCC-15 cells (1 × 10^4^ cells/well) were seeded in a 96-well plate. After 24 h of incubation at 37 °C in an incubator with a 5% CO_2_ atmosphere, the cells were treated with various concentrations of OA (0–500 μM) and CP (0–100 μM) for 24 h under the same incubation conditions. Then, the cells were incubated with 10 μL of CCK-8 reagent for 3 h at 37 °C. Absorbance was measured at 460 nm using the SpectraMax iD3 microplate reader. The assays were repeated three times independently.

### 2.8. Quantifying Apoptosis Using the Annexin V Dead Cell Kit

The apoptosis induced by KFE was quantified using the Annexin V Dead Cell Kit (Luminex, Austin, TX, USA) according to the manufacturer’s instructions. To prepare cell samples, 1.0 × 10^5^ cells/mL were seeded in a six-well plate and treated with various concentrations of KFE (12, 25, or 50 μg/mL) for 24 h at 37 °C in a 5% CO_2_ incubator. The cells were collected and stained with annexin V and 7-AAD reagent for 20 min at room temperature while shielded from light. Four populations were distinguished using the Guava Muse Cell Analyzer (Luminex).

### 2.9. Western Blot Analysis

SCC-15 and SCC-25 cells were treated with different doses of KFE. Harvested cells were lysed in PRO-PREP protein extraction solution (iNtRON Biotechnology, Seongnam, Republic of Korea). The protein concentration was measured using BCA reagent (Thermo Fisher Scientific, Waltham, MA, USA). A total of 30 μg of total protein was electrophoresed in 4–12% Bis-Tris gels (Thermo Fisher Scientific) and transferred to PVDF membranes (Millipore, Billerica, MA, USA) via electroblotting. The blots were incubated with blocking solution (5% skim milk) for 1 h, then probed with anti-microtubule-associated protein light chain 3 (*LC3*) (1:1000 dilution; Cell Signaling Technology, Danvers, MA, USA), anti-poly(ADP-ribose) polymerase (*PARP*) (1:1000 dilution; Cell Signaling), anti-p62/*SQSTM1* (1:1000 dilution; Cell Signaling), and anti-actin (1:1000 dilution; Santa Cruz Biotechnology, Santa Cruz, CA, USA) primary antibodies overnight at 4 °C. After washing, the blots were incubated with secondary antibodies (1:3000 dilution) for 1 h. The intensity of the protein bands was detected using a chemiluminescent substrate (Thermo Fisher Scientific). To quantify the band intensities, ImageJ software (NIH, Bethesda, MD, USA) was used.

### 2.10. Statistical Analysis

The data of the cell viability assay are expressed as means ± standard deviations (SDs) and the independent *t*-test was used for statistical analysis to compare the two groups. *p*-values were considered to be markedly significant at two different levels, *p* < 0.05 and *p* < 0.01. Each level of significance is marked as follows: * *p* < 0.05, ** *p* < 0.01. SPSS for Windows software (ver. 23.0; IBM Corp., Armonk, NY, USA) was used for the statistical analysis.

## 3. Results

### 3.1. Identification of KF Compounds and Their Targets

A search of four databases identified 40 compounds in KF after excluding duplicates (37 compounds in SymMap, 18 in TCMSP, 2 in HIT, and 36 in HERB) (Appendix A). The associations of the 40 compounds and source sites are depicted using a Venn diagram (Figure 1A). Compounds in KF were screened using the pharmacokinetic criteria of TCMSP and SwissADME (Table 1). When strictly applying the criteria of pharmacokinetic and physiological parameters, 5,7,4′-trihydroxy-6,3′-dimethoxyisoflavone, higenamine, hispidulin, isorhamnetin, and quercetin were suitable for Lipinski’s Ro5 with zero violation, and OB (%), caco-2, and DL values satisfied the applied criteria for active compounds. KFTs were collected from SymMap (372 genes), TCMSP (176 genes), HIT (11 genes), and HERB (40 genes). Overlapping gene sets are visualized using Metascape Circos plots (Figure 1B). After excluding redundancies, a list of 477 KFTs was generated (Appendix A).

### 3.2. Functions of KFTs Enriched in Biological Pathways

Metascape is an efficient and effective tool for analyzing and interpreting systems-level studies based on OMIC datasets, and simplifies the process of identifying enriched biological pathways. A total of 477 KFTs were uploaded to Metascape, and pathway and process enrichment analyses were performed. The Metascape bar graph summarizes the enriched biological functions, pathways, and processes associated with KFTs. The enriched functional categories are on the *y*-axis, and the enrichment score or *p*-value is on the *x*-axis. Each functional category is represented by a bar, and the height of the bar indicates the level of enrichment. The more enriched a functional category is, the higher its bar. The enriched terms were clustered hierarchically and 132 (27.7%) KF targets were in the ‘metabolism of lipids’ cluster (Figure 1C), while 81 (17%) were in the ‘pathways in cancer’ cluster. The ‘phosphatidylinositol 3′-kinase (PI3K)–AKT signaling pathway’ involved 45 (9.4%) KFTs (Figure 1C).

Targets with high interconnectivity were identified from the MCC score obtained using the cytoHubba plugin, and 10 hub genes were found: cellular tumor antigen p53 (*TP53*), vascular endothelial growth factor A (*VEGFA*), RAC-alpha serine/threonine-protein kinase (*AKT1*), serine/threonine-protein kinase mTOR (*MTOR*), phosphatidylinositol 3,4,5-trisphosphate 3-phosphatase and dual-specificity protein phosphatase PTEN (*PTEN*), caspase-3 (*CASP3*), myc proto-oncogene protein (*MYC*), G1/S-specific cyclin-D1(*CCND1*), and insulin (*INS*) (Figure 1D). Hub genes are genes that are highly connected to other genes within a biological network, and their central role in network communication and regulation makes them potentially important for various biological processes, including signal transduction, gene regulation, and disease pathways.

### 3.3. Relationships of KFTs with Autophagy, Apoptosis, and OSCC

A query of the STRING database (Cytoscape 3.9.1) yielded 88 OSCC-associated genes (confidence score cutoff = 0.40). Two 136-gene gene sets associated with autophagy and apoptosis were collected from the KEGG pathway database https://www.genome.jp/entry/hsa04140 (accessed on 11 February 2023), https://www.genome.jp/entry/hsa04210 (accessed on 11 February 2023). A FunRich 3.1.3 Venn diagram visualized common genes among the KF, OSCC, autophagy, and apoptosis gene sets (Figure 2). Three genes were common to all four groups: *AKT1*, apoptosis regulator bcl-2 (*BCL2*), and mitogen-activated protein kinase (*MAPK3*). *PTEN* and hypoxia-inducible factor 1 (*HIF1*) were involved in KF, OSCC, and autophagy. Eleven KFTs were involved in the autophagy pathway, and fifteen in the apoptosis pathway (Table 2 and Appendix A).

### 3.4. Analysis of the KF Compound OA: Function in Biological Processes and Pathways

The KF compound OA (Figure 3A) is a triterpenoid that has been isolated from >1000 plant species. OA met the MW, OB (%), Caco2 permeability, and DL criteria (Table 1). As shown in Figure 3B, a bioavailability radar plot created using SwissADME showed that lipophilicity and insolubility were slightly out of range. The pink area represents the suitable range for associated properties. OA targeted 29 genes (Figure 3C) and Metascape analysis indicated that the target genes were mainly involved in ‘response to hormones’, ‘lipids and atherosclerosis’, and ‘antiviral and anti-inflammatory effects of Nrf2 on SAR’ (Figure 3D). PPI analysis of OA targets generated 10 hub genes: *CASP3*, intercellular adhesion molecule 1 (ICAM1), nuclear factor erythroid-2-related factor 2 (*NFE2L2*), CASP8, heme oxygenase 1 (*HMOX1*), *RELA*, *CASP9*, receptor-type tyrosine-protein phosphatase C (*PTPRC*), dual-specificity protein phosphatase 3 (*DUSP3*), and NAD(P)H dehydrogenase [quinone] 1 (*NQO1*) according to the cytoHubba MCC score (Figure 3E).

### 3.5. Molecular Docking Analyses

Using AutoDock Vina, molecular docking was conducted for OA with six proteins. The RCSB PDB IDs of the proteins are: AKT1, 1UNP [33]; CASP3, 6BDV [34]; LC3, 1UGM [35]; SQSTM1, 6JM4 [36]; NFE2L2, 4IFL [37]; PTEN, and 1D5R [38]. The affinity scores for all docking results were <−6 kcal/mol. OA and NFE2L2 bonding had the strongest affinity score (–10.2 kcal/mol). The LigPlot for OA and the NFE2L2 complex showed an interaction with protein amino acids. The hydrogen bonds between OA and the glycine, valine, and isoleucine of NFE2L2 are shown in green along with the binding distances. By contrast, OA and CASP, as well as OA and PTEN, had −9.0 and −9.6 affinity, respectively, but docking mode did not indicate bonding and the LigPlot result showed no hydrogen bonds. OA and LC3 docking had an affinity score of −7.0, and OA interacted with the glutamic acid and threonine of LC3 via hydrogen bonds. OA and SQSTM1 docking had a binding affinity score of −7.2, and hydrogen bonds of SQSTM1 with phenylalanine and tyrosine were observed.

### 3.6. Effects of KFE, OA, and CP on the Viability of SCC-15 Cells: Results of MTT and CCK-8 Assays

According to the MTT assay, KFE (0–125 µg/mL) significantly inhibited SCC-15 cell proliferation. The IC₅₀ of KFE was 103.51 ± 2.07 μg/mL (Figure 4A). OA and CP were used at various doses to treat SCC-15 cells (1 × 10^4^/wells) in a 96-well plate. The SCC-15 cell viability on exposure to OA (Figure 4B) and CP (Figure 4C) was measured using the CCK-8 assay. The IC₅₀ of OA was 121.7 μM and that of CP was 25.47 ± 6.01 μM.

### 3.7. Mechanism Underlying SCC-15 Cell Death Induced by KFE Treatment

To assess the mechanism of SCC-15 cell death with KFE treatment, cells were stained with annexin V and 7-AAD. The assay indicated that the population of live cells in the control exceeded 89%. On increasing the KFE dose, the live cell proportions decreased to 62.24, 54.81, and 33.65% with 12, 25, and 50 μg/mL KFE, respectively. As shown in Figure 5, significant cell death occurred after KFE treatment; however, the cell population during early and late apoptosis rarely increased.

### 3.8. KFE and OA Induced Autophagic Cell Death in SCC-15 and SCC-25 Cells

SCC-15 cells were treated with KFE (40 μg/mL), OA (120 μM), and CP (15 μM) for 20 h. CP treatment, which is currently used as a first-line anticancer drug for head and neck cancer, resulted in PARP cleavage, an early event in apoptosis. By contrast, KFE and OA increased the protein levels of the autophagy markers p62/SQSTM1 and LC3 (Figure 6A). To confirm the contrasting results, an additional experiment was conducted on SCC-25 cells with the same method and dosages of KFE (40 μg/mL), OA (120 μM), and CP (15 μM) for 20 h, and comparable results were collected (Figure 6B).

## 4. Discussion

Using network-based systems pharmacology analysis, we identified 40 compounds (Table 1) and 477 KFTs (Appendix A). We were then able to distinguish 100 clusters through Metascape enrichment analysis and found that KFTs were most involved in ‘lipid metabolism’, followed by ‘protein phosphorylation’ and ‘cancer pathway’. ‘PI3K–AKT signaling pathway’ was among the top 20 clusters (Figure 1C). Among all tumorigenic signaling-promoting HNSCC pathways, PI3K–AKT–mTOR showed the most changes. We predicted strong associations between KF and OSCC cells, and hypothesized that KF could be used as an anti-tumor, anti-inflammatory, and anticancer agent. To test this, we further examined the data.

The MCC score of the cytoHubba plugin indicated 10 hub KFTs: *AKT1*, *CASP3*, *CCND1*, *MTOR*, *MYC*, *PTEN*, *TP53*, and *VEGFA* (Figure 1D). Genetic abnormalities such as oncogene activation and tumor suppressor gene degeneration play a role in tumor progression. HNSCC progression involves CDKN2A inactivation in the early stage of development, when normal mucosa becomes hyperplasic. *TP53* is inactivated during the subsequent progression to dysplasia. PTEN acts as a catalyst for the dephosphorylation of phosphatidylinosi-tol-3,4,5-triphosphate (*PIP3*). The loss of PTEN function increases PIP3 and activates the PI3K–AKT pathway, thereby promoting cancer cell growth and survival. Therefore, PTEN inactivation is involved in the in situ transition from dysplasia to carcinoma and invasive carcinoma [39].

OSCC was associated with 21 KFTs (AKT1, BCL2, CASP3, CASP8, CASP9, CCND1, CDK4, CDKN2A, ERBB2, ESR1, HIF1A, IL10, MAPK3, MET, MYC, PTEN, PTGS2, SLC2A1, TNF, TP53, and VEGFA), 11 of which were involved in autophagy (AKT1, TG4B, BCL2, CTSD, HIF1A, INS, MAPK3, PIK3R1, PRKACA, PTEN, and RAF1) and 15 in apoptosis (AKT1, BAX, BCL2, CASP3, CASP8, CASP9, CHUK, CTSD, MAPK3, PARP1, PIK3R1, RAF1, RELA, TNF, and TP53) (Table 2). Three KFTs (AKT1, BCL2, and MAPK3) were associated with OSCC, autophagy, and apoptosis. PTEN and HIF1 were involved in OSCC and autophagy (Figure 2). CASP3 plays a crucial role in inducing apoptosis. One KF compound, palmitic acid, targeted PTEN and BCL2 in the apoptosis pathway, and studies have reported that palmitic acid is associated with PTEN suppression and a slight reduction in BCL2 [40,41]. This indicates that palmitic acid is involved in cancer cell survival and apoptosis. However, the exact mechanism needs to be elucidated. ATG4 is an autophagy-regulating protease that cleaves LC3 into cytosolic LC3. Among the KF compounds analyzed in this study, n-triacontanol targeted ATG4.

Among the compounds in KF, the triterpenoid OA was selected for in-depth investigation, focusing on OSCC. This analysis revealed 29 targets of OA (Figure 3C) and the enriched Metascape analysis showed that OA target genes were involved in ‘antiviral and anti-inflammatory effects of Nrf2 on SAR’ (Figure 3D). PPI analysis of OA targets revealed 10 significant hub genes (*CASP3*, *CASP8*, *CASP9*, *DUSP3*, *HMOX1*, *ICAM1*, *NFE2L2*, *NQO1*, *PTPRC*, and *RELA*) according to cytoHubba MCC scores (Figure 3E). NFE2L2 encodes Nrf2, and low-dose OA was reported to have a hepatoprotective effect [42] and induced NFE2L2, which activates the transcription of antioxidant genes [43]. NFE2L2 is normally targeted by Keap1 for degradation. Under oxidative stress, the Nrf2 and Keap1 protein–protein interaction is disrupted by the stabilization of Keap1. Then, Nrf2 translocates to the nucleus, where it begins transcribing many genes including antioxidant enzymes that alleviate cell stress. NFE2L2 mutation is associated with poor outcomes in HNSCC. In this study, the molecular docking of OA with NFE2L2 showed strong binding affinity (–10.2 kcal/mol) and generated hydrogen bonds with glycine, valine, and isoleucine from NFE2L2 (Table 3).

To access the mechanism of action of OA on OSCC, molecular docking was performed. Among six receptor proteins (*AKT1*, *CASP3*, *LC3*, *NFE2L2*, *PTEN*, and *SQSTM1*), CASP and PTEN had a binding affinity of −9.6 kcal/mol, but no hydrogen bonds with OA were found. Molecular docking of OA and LC3 had a binding affinity of −7.0 kcal/mol, and there were interactions with threonine and glutamic acid. OA and SQSTM1 docking had a binding affinity of −7.2 kcal/mol and interaction with two amino acids of SQSTM1. OA is reported to regulate apoptosis by increasing p53, PARP, and CASP3 [44]. However, our docking results did not show an interaction between OA and CASP3. The molecular docking of OA with LC3 and SQSTM1 yielded high binding scores, and hydrogen binding with the receptor amino acids suggests that OA is involved in autophagy rather than apoptosis.

The anticancer effect was confirmed by treating SCC-15 cells with KFE, OA, and CP. KFE resulted in weak PARP cleavage, which is an early event in apoptosis, but strongly increased the levels of the autophagy marker proteins LC3 and p62/SQSTM1 (Figure 6). Interestingly, flow cytometry analysis using annexin V and 7-AAD revealed that KFE-induced apoptosis was quite rare (Figure 5). Further, in our experiments, OA rarely caused cell death; there were few changes in PARP, along with increases in p62/SQSTM1 and LC3 protein levels. This suggests that autophagy activation by OA inhibited apoptosis and contributed to SCC-15 cell growth. These results were consistent with the docking results. CP stimulated PARP cleavage, but did not change the LC3 or p62/SQSTM1 protein levels in SCC-15 cells (Figure 6). These results implied that KFE-induced SCC-15 cell death occurs via both apoptosis and autophagy pathways, but autophagy is significantly more important. Studies show that autophagy is often induced in cancer cells as a survival mechanism that improves their ability to grow; however, our results also imply that KFE-induced autophagy could be the main cause of SCC-15 cell death.

Overall, our laboratory findings imply that KF has anti-inflammatory, anti-tumor, and anticancer effects. However, before this natural product can be used as an adjuvant cancer treatment, real-world validation is required. Our study also showed that KFE can induce SCC-15 cell death via the autophagy signaling pathway. Modern cancer treatments remain limited and evidence of the efficacy of natural products, such as KF, as anticancer adjuvants is urgently needed to minimize patient suffering. Further research on this natural product could lead to advances in cancer treatment and anticancer pharmacology.

## 5. Conclusions

Network-based systems pharmacological analysis of KF identified 4 active compounds from among 40 compounds, and 477 KFTs. The KFTs were enriched in the ‘cancer pathway’ and ‘PI3K-AKT signaling pathway’. These results suggest that KF could have anticancer effects on OSCC. In addition, molecular docking analysis revealed that OA had hydrogen interactions with the autophagy-related marker proteins LC3 and SQSTM1, but not with CASP3. OA also showed high molecular affinity with NFE2L2, forming hydrogen bonds. The experimental validation results showed that the treatment of SCC-15 cells with KFE and OA increased autophagy marker protein levels, while flow cytometry revealed that KFE-induced apoptosis was relatively rare. In summary, KFE had significant cytotoxic effects on SCC-15 cells and induced autophagy. We hypothesize that KFE-induced autophagy may have a significant role in the death of SCC-15 cells. This study provides a basis for future investigations of new anti-OSCC bioactivity mechanisms specifically associated with autophagy. Further research on KF might lead to the discovery of possible adjuvants for advanced OSCC treatment and enhance modern pharmacology.

## Figures and Tables

**Figure 1 life-13-01300-f001:**
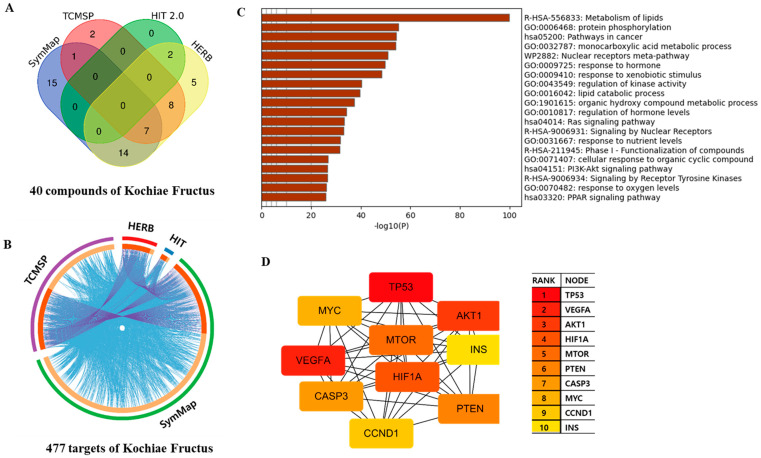
Results of network pharmacological and Metascape analyses of KF. (**A**) Venn diagram of the compounds found in KF and their sources. (**B**) Overlapping gene sets in a Circos plot obtained via a Metascape analysis. (**C**) Heatmap of the top 20 enriched clusters derived from a Metascape analysis of KFTs. Bar graph colors correspond to statistical significance; the lighter the color, the less significant the result. Metascape selects the term with the most significant statistical association (i.e., lowest *p*-value) within each cluster. Enriched cluster terms represent R-HAS: Reactome Gene Sets; GO: Gene Ontology biological processes; Has: KEGG Pathway; WP: Wiki Pathways. (**D**) Ten hub KFTs ranked by MCC score. MCC score is a measure of the quality of binary classifications, such as identifying a set of hub genes using cytoHubba. MCC score ranges from −1 to 1 (1: a perfect prediction, 0: a random prediction, −1: a perfectly wrong prediction).

**Figure 2 life-13-01300-f002:**
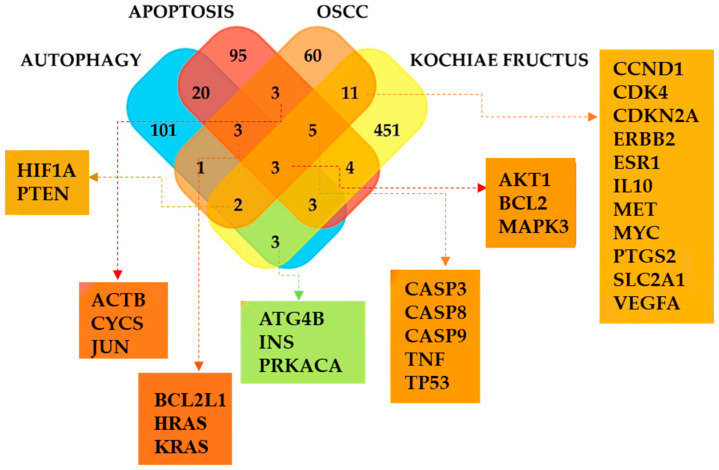
Genes related to KFTs, OSCC, autophagy, and apoptosis. Genes associated with KF, OSCC, autophagy, and apoptosis, and genes common to multiple datasets, were visualized using a FunRich Venn diagram. ACTB, Actin, cytoplasmic 1; ATG4B, Cysteine protease ATG4B; BCL2L2, Bcl-2-like protein 2; CASP8, Caspase-8; CASP9, Caspase-9; CDK4, Cyclin-dependent kinase 4; CDKN2A, Cyclin-dependent kinase inhibitor 2A; ERBB2, Receptor tyrosine-protein kinase erbB-2; ESR1, Estrogen receptor; HIF1A, Hypoxia-inducible factor 1-alpha; HRAS, GTPase HRas; IL10, Interleukin-10; JUN, Transcription factor Jun; MET, Hepatocyte growth factor receptor; PRKACA, mRNA of PKA Catalytic Subunit C-alpha; PTGS2, Prostaglandin G/H synthase 2; SLC2A1, Solute carrier family 2, facilitated glucose transporter member 1; TNF, Tumor necrosis factor.

**Figure 3 life-13-01300-f003:**
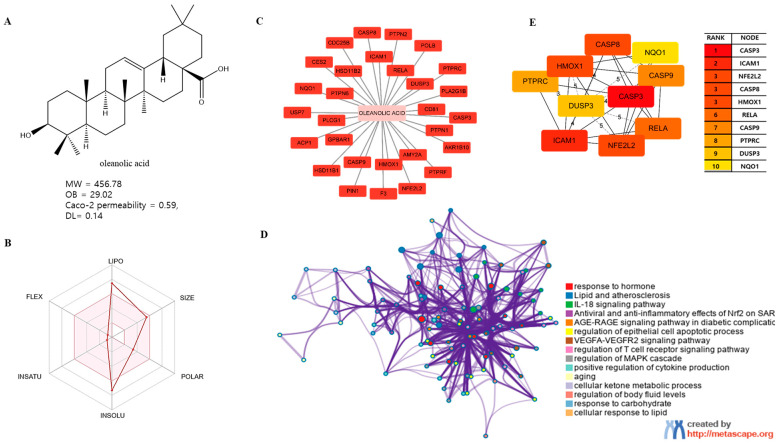
Network pharmacological and Metascape analyses of OA. (**A**) Structure of OA and ADME values. (**B**) SwissADME bioavailability radar plot of OA. LIPO (lipophilicity): −0.7 < XLOGP3 < +5.0; SIZE: 150 g/mol < MW < 500 g/mol; POLAR (polarity): 20 Å2 < TPSA < 130 Å2; INSOL (solubility): −6 < log S < 0; INSATU (saturation): 0.25 < Fraction of Csp3 < 1; FLEX (flexibility): 0 < number of rotatable bonds < 9. (**C**) Interactions of OA with its gene targets. (**D**) Metascape analysis of OA. (**E**) Significant OA targets according to the MCC score.

**Figure 4 life-13-01300-f004:**
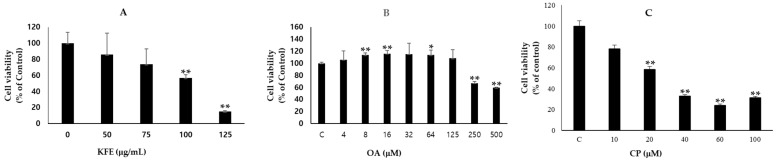
SCC-15 cell viability assay. Following treatment with KFE (**A**), OA (**B**), and CP (**C**), cell viability was determined using MTT assay for KFE, and the CCK-8 assay for OA and CP. The data are presented as mean ± S.D. of three independent experiments. Statistically significant differences were denoted as * *p* < 0.05 and ** *p* < 0.01 when compared with the corresponding control group (**C**).

**Figure 5 life-13-01300-f005:**
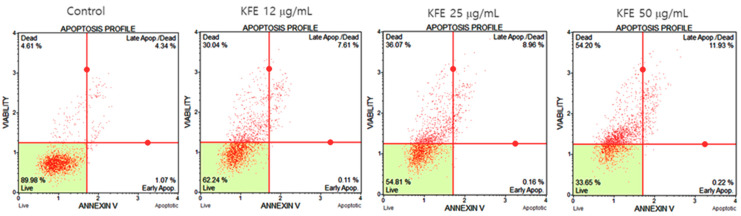
Annexin V and 7AAD staining assay was used to assess apoptotic SCC-15 cell death under KFE treatment. After a 16 h treatment with various doses and staining, cytometric analysis was performed using a MUSE^TM^ cell analyzer.

**Figure 6 life-13-01300-f006:**
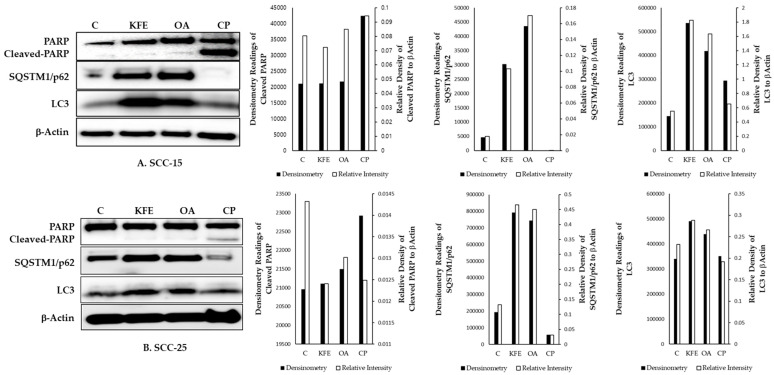
KFE and OA increased LC3 and p62/SQSTM1 protein levels, and cisplatin stimulated PARP cleavage in SCC-15 (**A**) and SCC-25 (**B**) squamous cell carcinoma cell lines. Expression levels of LC3, p62/SQSTM1, and PARP cleavage were determined by Western blotting after 20 h of treatment with KFE (40 μg/mL), OA (120 μM), and CP (15 μM) in SCC-15 (**A**) and SCC-25 (**B**) cells. β-Actin was used as a loading control. Full pictures of the Western blots are presented in Appendix A.

**Table 1 life-13-01300-t001:** Compounds of KF and Physiological and Pharmacological Characteristics.

Compounds	Formula	Physicochemical Parameters				Pharmacokinetic Parameters									
		MW	FASA	Hd	Ha	OB	Caco	BBB	Log	HL	GI	BA	DL	Lipinski	SA
(Z)-1,3-diphenylprop-2-en-1-one	C_15_H_12_O	208.3	0.53	0	1	47.27	1.48	1.18	3.7	20	High	0.6	0.08	Yes	2.41
11,14-Eicosadienoic Acid	C_20_H_36_O_2_	308.6	0.23	1	2	39.99	1.22	0.76	7.3	5.6	High	0.9	0.2	Yes	3.33
3-Acetylaconitine	C_36_H_49_NO_12_	687.9	0.2	2	13	37.05	−0.22	−0.73	−0.6	26	Low		0.2	No	7.64
^∗^ 5,7,4′-Trihydroxy-6,3′-Dimethoxyisoflavone	C_17_H_14_O_7_	330.3	0.22	3	7	63.36	0.54	−0.12	2.03	17	High	0.6	0.34	Yes	3.26
9, 12-Octadecadienoic Acid	C_39_H_70_O_6_	635	6	0	6	N/A	N/A	N	N	N	High	N	N	N	N
9E,12Z-Octadecadienoic Acid	C_18_H_32_O_2_	280.5	0.25	1	2	41.9	1.16	0.77	6.39	5.4	High	0.9	0.14	Yes	N
Aconitine	C_34_H_47_NO_11_	645.8	0.22	3	12	7.87	−0.58	−1.13	−1	N	Low	0.2	0.23	No	7.43
Arachidic acid	C_20_H_40_O_2_	312.6	0.18	1	2	16.66	1.18	1.09	8.19	N	Low	0.9	0.19	Yes	2.77
Benzoylaconine	C_32_H_45_NO_10_	603.8	0.21	4	11	12.83	−0.48	−1.12	−1.4	N	Low	0.2	0.25	No	7.16
Benzoylhypaconine	C_31_H_43_NO_9_	573.8	0.24	3	10	8.7	−0.29	−0.74	−0.5	N	High	0.6	0.29	Yes	6.95
Benzoylmesaconine	C_31_H_43_NO_10_	589.8	0.21	4	11	8.55	−0.52	−1.11	−1.7	N	Low	0.2	0.27	No	7.04
Daturic Acid	C_17_H_34_O_2_	270.5	0.21	1	2	18.51	1.12	0.95	6.82	N	High	0.9	0.12	Yes	2.42
Daucosterol	C_35_H_60_O_6_	577	0.23	4	6	20.63	−0.26	0.97	6.34	N	Low	0.6	0.63	Yes	8.02
Deoxyaconitine	C_34_H_47_NO_10_	629.8	0.2	2	11	30.96	−0.22	0.74	0.25	22	High	0.2	0.24	No	7.35
Diosgenin	C_27_H_42_O_3_	414.6	0.2	1	3	15.84	0.8	0.21	4.63	N	High	0.6	0.81	Yes	6.94
Docosanoate	C_22_H_43_O_2_-	340.7	0	1	2	15.69	1.21	0.91	9.11	N	High	0.9	0.26	Yes	2.96
Ecdysterone	C_27_H_44_O_7_	480.7	0.26	6	7	6.94	−1.38	1.93	1	N	High	0.6	0.82	Yes	6.36
Flavone	C_15_H_10_O_2_	222.2	0.46	0	2	25.88	1.34	1.08	3.14	N	High	0.6	0.13	Yes	2.88
^∗^ Higenamine	C_16_H_17_NO_3_	271.3	0.34	4	4	82.54	0.63	0.03	2.57	3.9	High	0.6	0.21	Yes	N
^∗^ Hispidulin	C_16_H_12_O_6_	300.3	0.28	3	6	30.97	0.48	0.49	2.32	16	High	0.6	0.27	Yes	3.12
Histamine	C_5_H_9_N_3_	111.2	0.09	3	2	44.1	0.5	0.37	−0.9	11	High	0.6	0.01	Yes	1.62
Hypaconitine	C_33_H_45_NO_10_	615.8	0.2	2	11	7.16	−0.1	0.41	−0.1	N	High	0.2	0.26	No	7.22
^∗^ Isorhamnetin	C_16_H_12_O_7_	316.3	0.32	4	7	49.6	0.31	0.54	1.76	14	High	0.6	0.31	Yes	3.26
Lignoceric Acid	C_24_H_48_O_2_	368.7	0.17	1	2	14.9	1.24	1.01	10	N	Low	0.9	0.33	Yes	3.24
Linoleic acid	C_18_H_32_O_2_	280.5	0.25	1	2	41.9	1.16	0.9	6.39	7.5	High	N/A	0.14	Yes	N
Linolenic Acid	C_18_H_30_O_2_	278.5	0.26	1	2	45.01	1.22	0.95	5.95	6.1	High	N/A	0.15	N	N
Mesaconitine	C_33_H_45_NO_11_	631.8	0.2	3	12	8.7	−0.35	0.95	−1.3	N	Low	0.2	0.24	No	7.31
Momordin Ic	C_41_H_64_O_13_	765.1	0.25	7	13	10.77	−1.69	2.23	3.65	N	Low	0.1	0.15	No	8.65
Myristic Acid	C_14_H_28_O_2_	228.4	0.19	1	2	21.18	1.07	0.99	5.46	N	High	0.9	0.07	Yes	2.09
Oleanolic Acid	C_30_H_48_O_3_	456.8	0.25	2	3	29.02	0.59	0.07	6.42	3.3	Low	0.9	0.76	Yes	6.08
Oleic Acid	C_18_H_34_O_2_	282.5	0.2	1	2	33.13	1.17	0.78	6.84	5	High	0.9	0.14	Yes	3.07
Palmitic Acid	C_16_H_32_O_2_	256.5	0	1	2	19.3	1.09	1	6.37	N	High	0.9	0.1	Yes	2.31
Phytol	C_20_H_40_O	296.6	0.22	1	1	33.82	1.23	0.85	7.34	2.3	Low	0.6	0.13	Yes	4.3
^∗^ Quercetin	C_15_H_10_O_7_	302.3	0.38	5	7	46.43	0.05	0.77	1.5	14	High	0.6	0.28	Yes	3.23
Saponin	C_58_H_94_O_27_	1223	0.29	13	27	1.74	−4.04	5.44	−1.4	N	Low	0.2	0.02	No	10
Stearic Acid	C_18_H_36_O_2_	284.5	0.19	1	2	17.83	1.15	1.22	7.28	N	High	0.9	0.14	Yes	2.54
Stigmasterol	C_29_H_48_O	412.8	0.22	1	1	43.83	1.44	1	7.64	5.6	Low	0.6	0.76	Yes	6.21
Triacontanol	C_30_H_62_O	438.9	0.14	1	1	11.19	1.42	0.7	12.8	N	Low	0.6	0.46	Yes	3.97
Yamogenin	C_27_H_42_O_3_	414.7	0.19	1	3	17.79	0.83	0.24	4.63	N	High	0.6	0.81	Yes	6.94
Zoomaric Acid	C_16_H_30_O_2_	254.5	0.24	1	2	35.78	1.18	0.88	5.92	5.3	High	0.9	0.1	Yes	2.84

MW: molecular weight; FASA: fractional negative accessible surface area; Hd: number of hydrogen bond donors; Ha: number of hydrogen bond acceptors; OB(%): oral bioavailability; Caco-2: Caco-2 permeability; BBB: blood–brain barrier; LogP: LogPo/w(ALogP); HL: half-life; GI: gastrointestinal absorption; BA: bioavailability score; DL: drug-likeness; Lipinski’s Ro5: Lipinski’s rule of five; SA: synthetic accessibility; N: not available; ∗: KF’s active compounds suitable for Lipinski’s rule of five criteria and ADME parameters.

**Table 2 life-13-01300-t002:** OSCC-, autophagy-, and apoptosis-associated genes.

Name of Gene Sets	Genes
KF & OSCC	*AKT1*, *BCL2*, *CASP3*, *CASP8*, *CASP9*, *CCND1*, *CDK4*, *CDKN2A*, *ERBB2*, *ESR1*, *HIF1A*, *L10*, *MAPK3*, *MET*, *MYC*, *PTEN*, *PTGS2*, *SLC2A1*, *TNF*, *TP53*, *VEGFA*
KF & Autophagy	*AKT1*, *ATG4B*, *BCL2*, *CTSD*, *HIF1A*, *INS*, *MAPK3*, *PIK3R1*, *PRKACA*, *PTEN*, *RAF1*
KF & Apoptosis	*AKT1*, *BAX*, *BCL2*, *CASP3*, *CASP8*, *CASP9*, *CHUK*, *CTSD*, *MAPK3*, *PARP1*, *PIK3R1*, *RAF1*, *RELA*, *TNF*, *TP53*

*CTSD*, Cathepsin D; *PIK3R1*, Phosphatidylinositol 3-kinase regulatory subunit alpha; *RAF1*, RAF proto-oncogene serine/threonine-protein kinase; *CHUK*, Inhibitor of nuclear factor kappa-B kinase subunit alpha; *PARP1*, Poly [ADP-ribose] polymerase 1; *RELA*, Transcription factor p65.

**Table 3 life-13-01300-t003:** Molecular docking of OA and proteins.

Proteins	RCSB PDB ID	Affinity (kcal/mol)	Molecular Docking Modes	Ligand–Protein Interaction
AKT1	1UNP	−8.1	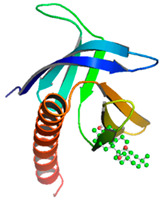	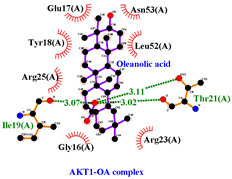
CASP3	6BDV	−9.0	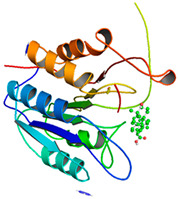	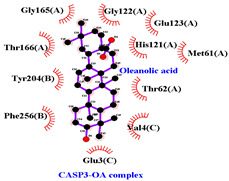
LC3	1UGM	−7.0	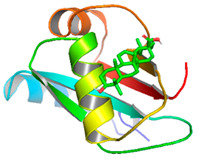	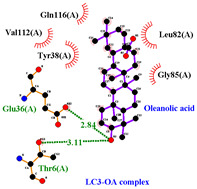
SQSTM1 (p62)	6JM4	−7.2	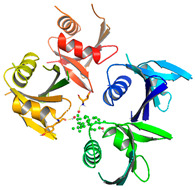	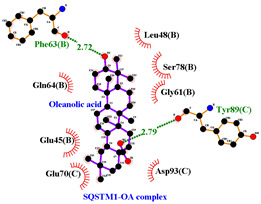
NFE2L2	4IFL	−10.2	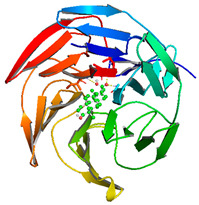	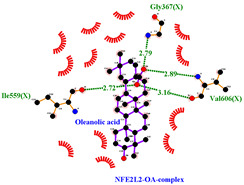
PTEN	1D5R	−9.6	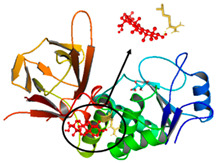	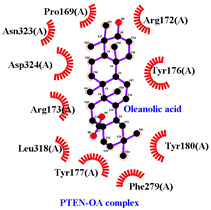

## Data Availability

The data used to support the findings of this study are included within the article.

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
