# Peer review of "Discovering Potential Anti-Oral Squamous Cell Carcinoma Mechanisms from Kochiae Fructus Using Network-Based Pharmacology Analysis and Experimental Validation"

_life, 2023, doi:10.3390/life13061300_

Round 1
Reviewer 1 Report
Dear authors,
The article. " Discovering Potential Anti-Oral Squamous Cell Carcinoma Mechanisms from Kochiae Fructus via Network-based Pharmacology Analysis, and Experimental Validation." suggests that Kochiae Fructus (KF), a natural product considered one of the herbal medicines, exhibits anticancer effects by inducing autophagy against oral cancer cell lines.
Chemotherapy is often the treatment of choice as a therapeutic approach for oral cancer. Anticancer drugs are often used as adjuvant therapy, especially for patients at high risk of recurrence and metastasis. However, anticancer drugs often have strong side effects and cannot be used for patients in poor general condition or cannot be used for long periods of time, and there is an urgent need to improve these problems. The KF used in this paper is a natural product and is considered to be relatively harmless to living organisms; if KF and other natural products can be selected as anticancer agents, it could expand treatment options and is a very interesting theme.
However, this research needs to be improved before the paper is published.
Major comments:
I believe the goal of this paper is to discover the biological actions of KF components and their targets using network analysis. Although network analysis was used to predict the binding of KF compounds to multiple cancer-related proteins, only p62 and LC3 have actually been verified in cell experiments with expression changes. p62 and LC3 were predicted to bind to proteins such as AKT and CASP3, yet only these two have confirmed expression changes What is the reason? Also, several apoptosis-related genes/proteins are predicted as targets of KF, but I feel that there is a discrepancy with the analysis because apoptosis is not occurring.
Also, there does not appear to be a strong relationship between KF components and autophagy in the network analysis. If you want to show a link to autophagy, you need experiments that show that autophagy is actually occurring in SCC-15.
Minor comments:
â‘ Suddenly the mention of KFE appears in the Abstract; there is a mention of KF extract (KFE) in the Introduction, but I would suggest you mention it this way in the Abstract. Also, perhaps you abbreviate Autophagy as AUP, is this a common abbreviation? I think it would be easier to understand if you first mention Autophagy (AUP).
â‘¡Lines 46-53 of the Introduction mention biomarkers, but this paper does not search for biomarkers, and the story seems to be developing in a less relevant way. Also, the next sentence that talks about cisplatin reads like it is commonly used as a radiosensitizer, but cisplatin is also associated with apoptosis, and I would recommend a more detailed description of cisplatin's role as an anticancer agent.
â‘¢In the Introduction, there is a statement that it has not yet been confirmed whether cisplatin exhibits dose-dependent effects, please provide a reference. Also, there is a statement that there are growing expectations for the role of natural products as anti-cancer agents and that there are unlimited herbal drug resources for research, please provide references for this as well.
â‘£In the Introduction, we recommend that you note the details of the network analysis that is the main focus of this paper.
⑤Fig.1 A.B is a picture of a crude drug, is it necessary as a figure? What is the significance of including them?
â‘¥We recommend that you describe your network analysis method a little more clearly.
⑦Although only SCC-15 was used in this study, it is recommended that multiple cell lines be considered.
⑧Please tell us why you use CCK-8 and MTT in some cases to look at cell viability and in others you use MTT.
⑨Table 1 and Table S1 say that 14 different KF compounds were screened, but in Table S1 there are 36 data, what does this tell us?
â‘©What does the log on the horizontal axis in Figure 1E represent? I recommend adding a little more explanation on how to see Figure 1D.
⑪In result 3.7, there is a statement that treatment with KFE caused significant cell death as shown in Figure 2, but is this a mistake in Figure 5?
â‘«In the MTT assay, it seems that the survival rate when KFE is added is significantly different at concentrations of 100 μg/ml or higher, but why was the concentration of KFE used below 50 μg/ml in the cell death experiment using Annexin V? Also, please explain why the concentration of CP in the experiment in Result 3.8 was 15 μM, whereas the concentrations of KFE and OA were 40 μg/ml and 120 μM, respectively, which were not significantly different in the survival experiment using CCK-8.
⑬The discussion states that KFT is second involved in the "cancer pathway," but the figure suggests that "protein phosphorylation" is second and "cancer pathway" is third, but since the difference is minute, is the interpretation that "protein phosphorylation," "cancer pathway," and "carboxylic acid metabolism" are all second involved?
Author Response
"Please see the attachment."

Reviewer 2 Report
Oral squamous cell carcinoma (OSCC) is the most common oral cancer, presenting in 90% of cases. It usually localizes at the vermilion border of the lower lip, the floor of the mouth and lateral border of the tongue. Treatment at the early stages of OSCC consists of surgical resection or radiotherapy.
Youn-Sook Kim and colleagues report in their manuscript titled “ Discovering Potential Anti-Oral Squamous Cell Carcinoma Mechanisms from Kochiae Fructus via Network-based Pharmacology Analysis, and Experimental Validation.” using network-based systems pharmacological analysis and docking analysis, predicted high associations between, Kochiae Fructus (KF) and OSCC cells and anticipated its possible use as an anti-cancer agent.
Additionally, in vitro studies using cell line SCC-15 have shown that the KF extract and a single compound - oleanonic acid caused SCC-15 cell deaths and induced accumulation of autophagy marker protein levels (LC3 and p62/SQSTM). Authors suggest that the KF extract maybe will be new therapeutic candidate for the treatment of patients with OSCC. |
The conception, execution and interpretation of the experiments in their current form are not very convincing, but they need to be refined.
Major points:
1. In vitro study results based on only one cell line SCC-15 (single donor)
2. There was no assessment of the influence of the KF extract on the expression level of the 10 hub genes (for example selected 3 or 4 genes: TP53 or AKT1, MYC PTEN)
Minor pointes
3. I recommend the unify the correct format of human gene symbols in the whole article. The symbols of the human genes are written in italics and in capital letters.
Author Response
"Please see the attachment."

Reviewer 3 Report
Comments:
1. The abstract should highlight, with concrete data, what the presented research study brings and not what the objectives were.
2. Line 24: KFE should change to KF extract
3. Section 1 -Introduction: should be improved considering the large number of works dealing with the subject of active biological compounds existing in Kochiae Fructus.
4. Line 98: The program to visualize the 3D docking poses is Pymol (not Phymol). Please revise.
5. Line 150: should change to “SCC-15 cell viability following treatment with KFE was measured….”. (the abbreviation of KFE was used in the previous section).
6. Section 3.4: Swiss ADME was used to understand the pharmacokinetics and ADMET properties of compounds. In addition, orally active drugs should obey Lipinski rule of five and Jorgensen rule of three. Please discuss this in this section.
7. Line 272: “ATK1: 1unp” should change to “ATK1: 1UNP”. Also, references of proteins used in this section are required.
8. Table 3: The figures of 2D interaction need to improve the resolution. The reader cannot see the amino acid residues name in this table.
9. Section 3.6: “IC50” ----- “IC50”
10. Compounds reported in Table 1 are previously reported in many medicinal plants. Oleanonic acid, a known compound, was reported in more than 500 published papers and easily obtained from commercial sources. So what is the difference between KF with another medicinal plants, and what is the novelty of this study? Please discuss this in Section 4 - Discussion.
Author Response
"Please see the attachment."

Reviewer 4 Report
The manuscript by Sung and colleagues describes the in vitro effect of Kochiae Fructus extract (KFE), as well as of its components, on Squamous Carcinoma Cells (SCC-15). The latter, exposed to KFE, upregulatedthe expression of protein markers of autophagy.
Several issues need to be solved. Further details are reported below.
- The novelty of the study should be calrified. Indeed, the Introduction section does not provide enough background information on the Kochiae Fructus properties for cancer treatment. If it is already known for the anticancer properties, as cited in ref. 7 by the Authors, what is new in this paper? Why the Authors chose to investigate KF impact on cells SCC-15? Are there other studies in literature claiming particular effects of KF in oral cancer treatment? Are there other works reporting the effects of KF and autophagy?
- The Materials and methods section did not explain, or provide references, to support the choice of methanol to prepare KF extract. Why not ethanol, or water? Similarly, the dissolution of the lyophilized powder in DMSO for in vitro experiments is not commented. Furthermore, the yield of the extraction procedure is very low (4.45%).
- English language should be carefully revised throughout the manuscript.
Author Response
"Please see the attachment."

Round 2
Reviewer 1 Report
Dear authors,
Thank you for your kind response to my comments.
The authors have responded adequately to some reviewer suggestions.
However, there are a few more points I would like to confirm. Are you sure that you do not mean research on autophagy as a mechanism for keeping cells alive, but on autophagic cell death? Sorry, I was misunderstood.I think readers will be very confused if it is described as autophagy.
Major comments
You state in your comment that ' our findings suggest that KF can inhibit apoptosis by inducing autophagy in SCC-15 cells.', but isn't the general perception that autophagy cell death is a backup function that is triggered instead when apoptosis is not induced? Isn't it too much of a leap to say that autophagy suppresses apoptosis based on these results?
Sorry for the lack of clarity. My point about showing that autophagy is occurring is not the result of Western blotting, but that the phenomenon of autophagy should be confirmed by electron microscopy or a commercially available kit.
Also, if you have a lot of results, why not add them to the RESULT or SUPPLEMENT as a data augmentation? Those looking at changes in phosphorylation need results for unphosphorylated protein as a control.
I think it would increase the reliability of your results if you also included the results of the SCC-25 experiments in your paper in important areas, if not all. (Minor comment⑦)
We know the difference between the MTT assay and the WST of CCK-8. We do not want you to explain the difference between the methods to us. We are recommending that if you are comparing the effects of drugs, you should show the results of the assessment using the same method rather than showing the results of methods with different cytotoxicity and sensitivity. (Minor comment⑧)
It is understandable that changes in relevant proteins may occur prior to cell death, but there is too much difference in cell viability between the results in Fig. 4 and Fig. 5. Also, the distribution of cells seems to be less common in this experiment. Is there any influence of the cell condition or experimental technique rather than the drug? And Fig. 6 shows another experiment with a different concentration. While it is important to find the best conditions for clean results, it is also important to conduct experiments under uniform conditions. If it is stated that the effect is dose-dependent, but the results change with slightly different concentrations, it is unreliable. It is recommended to state the same concentration results to show reliability. (Minor comment⑫)
Reviewer 2 Report
I am satisfied with the corrections and additional experiments made by the authors
Author Response
We have received your additional comment through the Academic Editor. The requested adjustments have been applied on our manuscript and have been sent to the Academic Editor.
We greatly apprecieate your review. Thank you for your positive comments.
Reviewer 3 Report
The authors have satisfactorily responded to all comments raised by the reviewer. I am pleased to recommend the acceptance of this manuscript to Life.
Author Response
We greatly appreciate your valuable review. Thank you for your recommendation.
Reviewer 4 Report
The revised manuscript has been revised and all the suggested editing has been performed.
Author Response
We appreciate your time on reviewing our manuscript.